# Intravenous versus inhalational maintenance of anesthesia for quality of recovery in adult patients undergoing non-cardiac surgery: A systematic review with meta-analysis and trial sequential analysis

**Min Shui[☉], Ziyi Xue[☉], Xiaolei Miao, Changwei Wei\*, Anshi Wu\***

Department of Anesthesiology, Beijing Chaoyang Hospital, Capital Medical University, Beijing, China

☉ These authors contributed equally to this work.
\* changwei.wei@ccmu.edu.cn (CW); wuanshi88cy@163.com (AW)

## Abstract

### Background

Intravenous and inhalational agents are commonly used in general anesthesia. However, it is still controversial which technique is superior for the quality of postoperative recovery. This meta-analysis aimed at comparing impact of total intravenous anesthesia (TIVA) versus inhalational maintenance of anesthesia on the quality of recovery in patients undergoing non-cardiac surgery.

### Methods

We systematically searched EMBASE, PubMed, and Cochrane library for randomized controlled trials (RCTs), with no language or publication status restriction. Two authors independently performed data extraction and assessed risk of bias. The outcomes were expressed as mean difference (MD) with 95% confidence interval (CI) based on a random-effect model. We performed trial sequential analysis (TSA) for total QoR-40 scores and calculated the required information size (RIS) to correct the increased type I error.

### Results

A total of 156 records were identified, and 9 RCTs consisting of 922 patients were reviewed and included in the meta-analysis. It revealed a significant increase in total QoR-40 score on the day of surgery with TIVA (MD, 5.91 points; 95% CI, 2.14 to 9.68 points; $P = 0.002$; $I^2 = 0.0\%$). The main improvement was in four dimensions, including "physical comfort", "emotional status", "psychological support" and "physical independence". There was no significant difference between groups in total QoR-40 score ($P = 0.120$) or scores of each dimension on POD1. The TSA showed that the estimated required information size for total QoR-40 scores was not surpassed by recovered evidence in our meta-analysis. And the

**Data Availability Statement:** All relevant data are within the manuscript and its Supporting Information files.

**Funding:** The authors received no specific funding for this work.

**Competing interests:** The authors have declared that no competing interests exist.

adjusted Z-curves did not cross the conventional boundary and the TSA monitoring boundary.

## Conclusion

Low-certainty evidence suggests that propofol-based TIVA may improve the QoR-40 score on the day of surgery. But more evidence is needed for a firm conclusion and clinical significance.

## 1. Introduction

The requirement for effectiveness and efficiency of healthcare resources prompts anesthesiologists to consider techniques that provide a fast and high quality of recovery. However, various factors impact quality of recovery of patients after surgery. These factors may lead to prolonged surgical recovery, delayed hospital discharge and increased cost.

Intravenous and inhalational agents are commonly used in general anesthesia. Several comparisons between the two anesthesia techniques have been conducted previously. Research has focused on the impact of total intravenous anesthesia (TIVA) versus inhalational anesthesia maintenance on outcomes such as emergence time, perioperative hemodynamic parameters, pain scores and analgesic consumption, postoperative nausea and vomiting (PONV), length of hospital stay and other adverse events [1–4]. There were also several meta-analyses on some of these topics [1,5,6]. In recent years, researchers have recognized the limitations of the fragmentary measures and started to focus on global assessment of recovery quality from different perspective [7].

The Quality of Recovery-40 (QoR-40) questionnaire is an extensively validated instrument to assess the quality of recovery after surgery and anesthesia [7,8]. The QoR-40 questionnaire is composed of 40 items and incorporates five dimensions of health: emotional state, physical comfort, psychological support, physical independence and pain. Each item is rated on a five-point Likert scale: none of the time, some of the time, usually, most of the time, and all of the time. The total score of QoR-40 questionnaire ranges from 40 (poorest quality of recovery) to 200 (best quality of recovery) [7,8].

Several studies reported that different anesthesia techniques affected the quality of postoperative recovery. But the results of these studies remain controversial. Some reported no significant statistical difference in QoR-40 score between the two anesthesia techniques [9–12]. Some reported that TIVA group had higher QoR-40 score [3,13]. Currently, there are no published systematic review or meta-analysis on this subject. We hypothesized that TIVA could improve the quality of postoperative recovery. This systematic review and meta-analysis aimed to identify the impact of TIVA compared to inhalational maintenance of anesthesia on the quality of recovery in adult patients undergoing non-cardiac surgery.

## 2. Materials and methods

This systematic review and meta-analysis adhered to the recommendations of the Cochrane Collaboration [14] and is reported per Preferred Reporting Items for Systematic Reviews and Meta-Analysis (PRISMA) guidelines [15]. The protocol was registered in the international prospective register of systematic reviews (PROSPERO) (CRD42020188757).

## 2.1. Search strategy and study selection

Three databases (Excerpta Medica database (EMBASE), PubMed, The Cochrane Central Register of Controlled Trials (CENTRAL)) were systematically searched for relevant trials from inception to June 19, 2020 with no language or publication status restriction. We also manually checked the reference lists of relevant papers to identify additional studies. Search terms included "anesthesia, intravenous", "anesthesia, inhalation", "anesthetics, inhalation", "balanced anesthesia" and "quality of recovery" (S1 File).

We only included randomized controlled trials (RCTs) that compared the impact of propofol-based TIVA versus inhalational maintenance of anesthesia on the quality of recovery using the QoR-40. Inclusion criteria included: (1) Population: adult patients (age > 18 years) undergoing non-cardiac surgery under general anesthesia; (2) Intervention: propofol-based TIVA; (3) Comparator: inhalational maintenance of anesthesia; (4) Outcomes: (primary outcome) the total QoR-40 scores at different time point; scores of five dimensions, including physical comfort, emotional state, physical independence, psychological support and pain; (5) Study design: RCTs published in full-text versions. Studies with assessment tool for postoperative recovery quality other than QoR-40 were excluded.

## 2.2. Data extraction

We used reference management software (Medref 5.0) to collate the results of the searches and remove duplicates. Two authors (M.S. and Z.Y.X.) independently screened the results from titles and abstracts, and identified potentially relevant studies according to the eligibility criteria. The two authors independently performed reviews of full-text articles and data extraction. Any discrepancies were resolved by discussing with a third author (C.W.W.). We extracted information including study characteristics (publication year, design, setting and outcomes), participants (demographic characteristics, sample size and type of surgery), experimental intervention (induction technique, type of anesthetics, use of depth of anesthesia monitoring, administration regimen) and outcomes (total QoR-40 score, scores of each dimension, and time point). Means and standard deviations (SDs) were extracted from each study. If the data were normally distributed, median values and interquartile ranges were converted to means and SDs (http://www.math.hkbu.edu.hk/~tongt/papers/median2mean.html). We obtained estimated values from graphs or figures where numeric scores were not reported (https://apps.automeris.io/wpd/index.zh_CN.html). We contacted authors when information or results from the articles was insufficient.

## 2.3. Risk of bias and quality of evidence assessment

Two authors (M.S. and Z.Y.X.) independently assessed risk of bias using RevMan5.4 software, which assesses the following seven domains: Random sequence generation, allocation concealment, blinding of participants and personnel, blinding of outcome assessment, incomplete outcome data, selective reporting, and other bias. According to the Cochrane tool, each domain was classified as having low, unclear, or high risk of bias. We assessed publication bias through visual inspection of the funnel plot because of small number of included studies. The quality of evidence was classified using the GRADEpro software as high, moderate, low, or very low for each outcome based on five domains (the risk of bias, inconsistency, indirectness, imprecision, and publication bias) [16,17].

## 2.4. Statistical analyses

We used Review Manager 5.4 software for statistical analysis. All of the outcomes were quantitative variables and expressed as mean difference (MD) and standard deviation (SD) with

respective 95% confidence interval (CI) based on a random-effect model (DL). A wide CI revealed imprecision in our results. Forest plots were used to illustrate the estimates of overall effects. *P* value < .05 was considered as statistically significant difference. Subgroup analysis for the outcomes was based on the time point of questionnaire assessment. Heterogeneity of each outcome was assessed by $I^2$ statistic and the Chi-square test. $I^2 < 25\%$, $25\% \leq I^2 < 50\%$, $50\% \leq I^2 < 75\%$, and $I^2 \geq 75\%$ were considered as nil, mild, moderate and strong heterogeneity, respectively. In addition, we considered the point estimates and the overlap of CIs. We performed sensitivity analysis using a fixed-effect model. We also excluded each study sequentially to assess the impact of individual trials on the overall effect estimates. We performed trial sequential analysis (TSA) for total QoR-40 scores and calculated the required information size (RIS) to correct the increased type I error. The overall type I risk was set as 5%, with a power of 80%.

## 3. Results

### 3.1. Search results

A total of 156 articles were identified through the databases and other sources. After removing duplicate records, we screened the remaining 144 relevant publications. We obtained 19 full-text reports to assess eligibility (Fig 1) and excluded 10 references (3 conference paper, 1 duplicate data, 1 inadequate study design, 1 different intervention, 1 outcome using QoR-15 questionnaire, 1 lack of enough information, 2 ongoing studies). Finally, 9 randomized controlled trials (RCTs) consisting of 922 patients (462 in the TIVA groups and 460 in the inhalational maintenance groups) were reviewed [3,9–13,18–20], and data from all of these were included in meta-analysis.

### 3.2. Study characteristics

Of the included studies, 3 RCTs were performed in gynecological laparoscopies, 2 in otorhinolaryngology surgery, 1 in laparoscopic cholecystectomy, 1in thyroid surgery, 1 in vitrectomy and 1 in rhytidoplasty, respectively. All studies compared total intravenous anesthesia (TIVA) using propofol versus maintenance anesthesia using inhalational agents. All studies used intravenous agents during anesthesia induction in both groups. Four studies described propofol anesthesia using target-controlled infusion (TCI) [3,12,13,20]. Five studies compared TIVA versus maintenance using sevoflurane [9–12,18]. Three studies compared TIVA versus maintenance using desflurane [3,13,20]. One study did not report details of the inhalational agent [19]. Remifentanil was used in both groups during anesthesia maintenance except one study using ketamine in TIVA group [19]. Seven studies described monitoring of depth of anesthesia in both groups using of bispectral index (BIS). One study [19] described BIS monitoring only in TIVA group and the information was not available in another study (Table 1) [9].

### 3.3. Risk of bias assessment

Overall, most studies were considered as low risk of bias (Fig 2). Only one study did not describe the details of random sequence generation [19]. Details of allocation concealment were not available in three studies [3,19,20]. There was no difference between the two reviewers (M.S. and Z.Y. X.) for assessment of risk of bias in any study.

### 3.4. Outcomes

Nine studies reported total QoR-40 scores. The questionnaire was assessed in postanesthesia care unit (PACU) in one study [19], two at 6 hours after surgery [3,20], five at 24 hours after

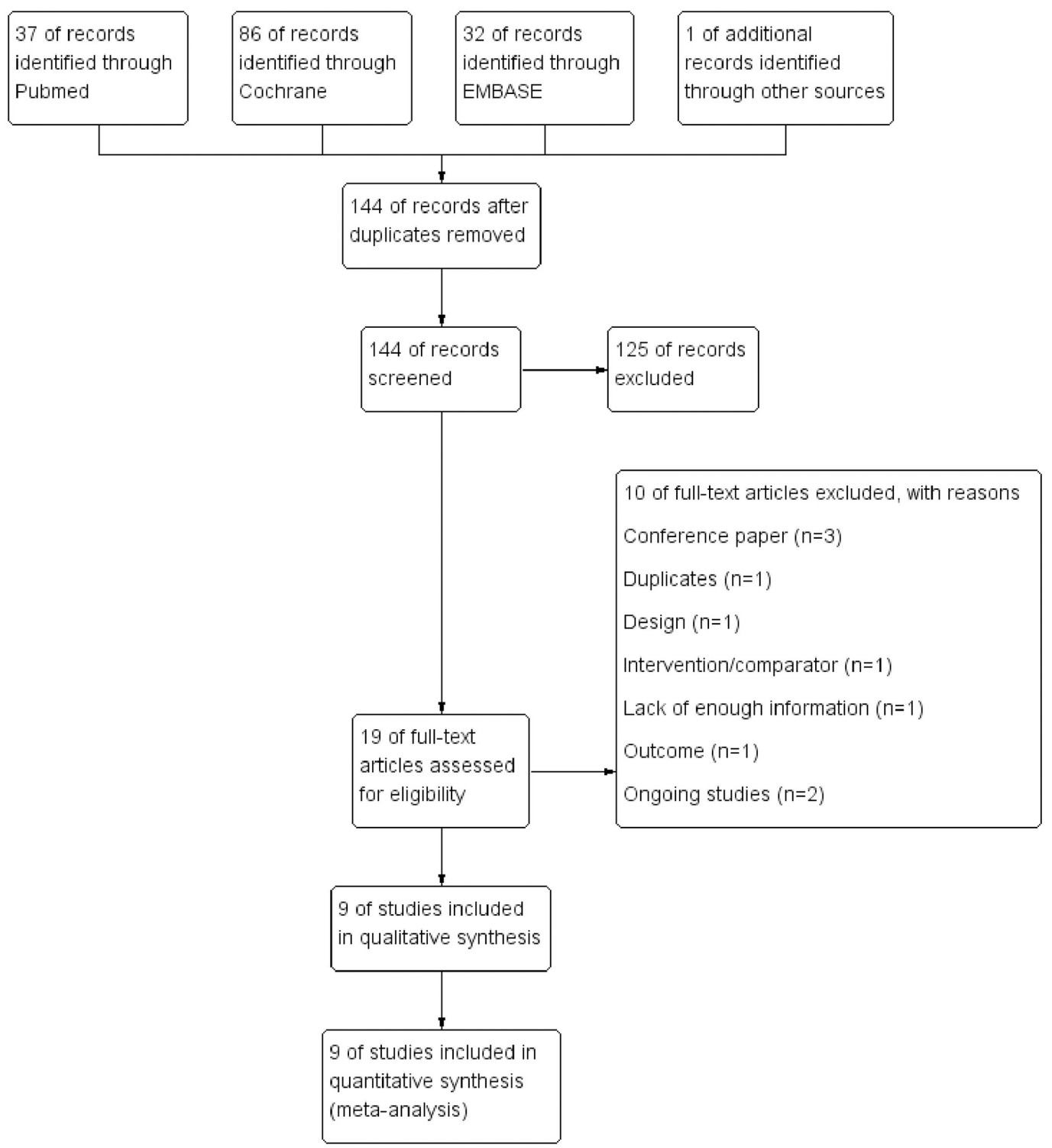

**Fig 1. Study flow diagram.**

**Table 1. Characteristics of included studies.**

| Trial | Country | Setting | ASA status | Age: (yr ± SD) | Gender: M/ F | Sample size | BIS monitoring | Surgery | Anesthesia | | Outcomes |
|---|---|---|---|---|---|---|---|---|---|---|---|
| | | | | | | | | | Inhalation | TIVA | |
| Li (2012) [18] | China | Inpatient | I, II | TIVA: 27.8 ±3.9 Inhalation: 28.2±4.1 | TIVA: 0/30 Inhalation: 0/30 | TIVA: 30 Inhalation: 30 | Yes | Gynecological laparoscopies | Sevo + Remi | Prop + Remi | Total QoR-40 score at 24h after surgery |
| Mei (2014) [10] | China | Inpatient | I, II | TIVA: 28.6 ± 4.8 Inhalation: 28.1 ± 4.2 | TIVA: 0/ 147 Inhalation: 0/148 | TIVA:147 Inhalation: 148 | Yes | Gynecological laparoscopies | Sevo + Remi | Prop + Remi | Total QoR-40 score at 24h after surgery |
| Jones (2015) [19] | USA | Outpatient | NR | Overall: 63.5 ± 6.8 | TIVA: 0/15 Inhalation: 0/15 | TIVA:15 Inhalation: 15 | Only in TIVA group | Rhytidoplasty | NR | Prop + Ketamine | Total QoR-40 scores at PACU, and on POD1 |
| Lee (2015) [13] | Korea | Inpatient | I, II | TIVA: 43.6 (23–60) Inhalation: 40.0 (25–61) | TIVA: 0/38 Inhalation: 0/38 | TIVA: 38 Inhalation: 38 | Yes | Thyroid surgery for neoplasm | Des + Remi | Prop (TCI) + Remi (TCI) | Total and each dimension scores of QoR-40 on POD1 and POD2 |
| Moro (2016) [9] | Brazil | Inpatient | I, II | TIVA: 37.9 ± 11.5 Inhalation: 39.3 ± 12.7 | TIVA: 21/ 33 Inhalation: 25/31 | TIVA: 54 Inhalation: 56 | NR | Otorhinolaryngological surgery | Sevo + Remi | Prop +Remi | Total and each dimension scores of QoR-40 at 24h after surgery |
| De Oliveira (2017) [11] | USA | Outpatient | I, II | TIVA: 40.1 ± 11.5 Inhalation: 39.2 ± 12.4 | TIVA: 0/37 Inhalation: 0/30 | TIVA: 37 Inhalation: 30 | Yes | Gynecological laparoscopy | Sevo + Remi | Prop +Remi | Total and each dimension scores of QoR-40 at 24h after surgery |
| Na (2018) [20] | Korea | Outpatient | I, II, III | TIVA: 59.0 (51.0–64.0) Inhalation: 60.0(50.0–70.0) | TIVA: 17/ 24 Inhalation: 19/23 | TIVA: 41 Inhalation: 42 | Yes | Vitrectomy | Des + Remi (TCI) | Prop (TCI) + Remi (TCI) | Total and each dimension scores of QoR-40 at 6h after surgery |
| Liu (2019) [3] | China | Inpatient | I, II | TIVA: 41.1 ± 12.6 Inhalation: 44.8 ± 1.43 | TIVA: 25/ 15 Inhalation: 27/13 | TIVA: 40 Inhalation: 40 | Yes | Endoscopic sinus surgery | Des + Remi | Prop (TCI) + Remi | Total and each dimension scores of QoR-40 at 6h after surgery, and on POD1 |
| De Carli (2020) [12] | Brazil | Inpatient | I, II | TIVA: 43.78±13.24 Inhalation: 43.95±11.43 | TIVA: 0/60 Inhalation: 0/61 | TIVA: 60 Inhalation: 61 | Yes | Laparoscopic cholecystectomy | Sevo + Remi (TCI) | Prop (TCI) + Remi (TCI) | Total and each dimension scores of QoR-40 at 24h after surgery |

Abbreviation: ASA, American Society of Anesthesiologists; BIS, bispectral index; Des, desflurane; NR, not reported; PACU, postanesthesia care unit; POD1, postoperative day 1; POD2, postoperative day 2; QoR-40, Quality of Recovery-40 questionnaire; Prop, propofol; Remi, remifentanil; Sevo, sevoflurane; SD, standard deviation; TCI, target-controlled infusion; TIVA, total intravenous anesthesia.

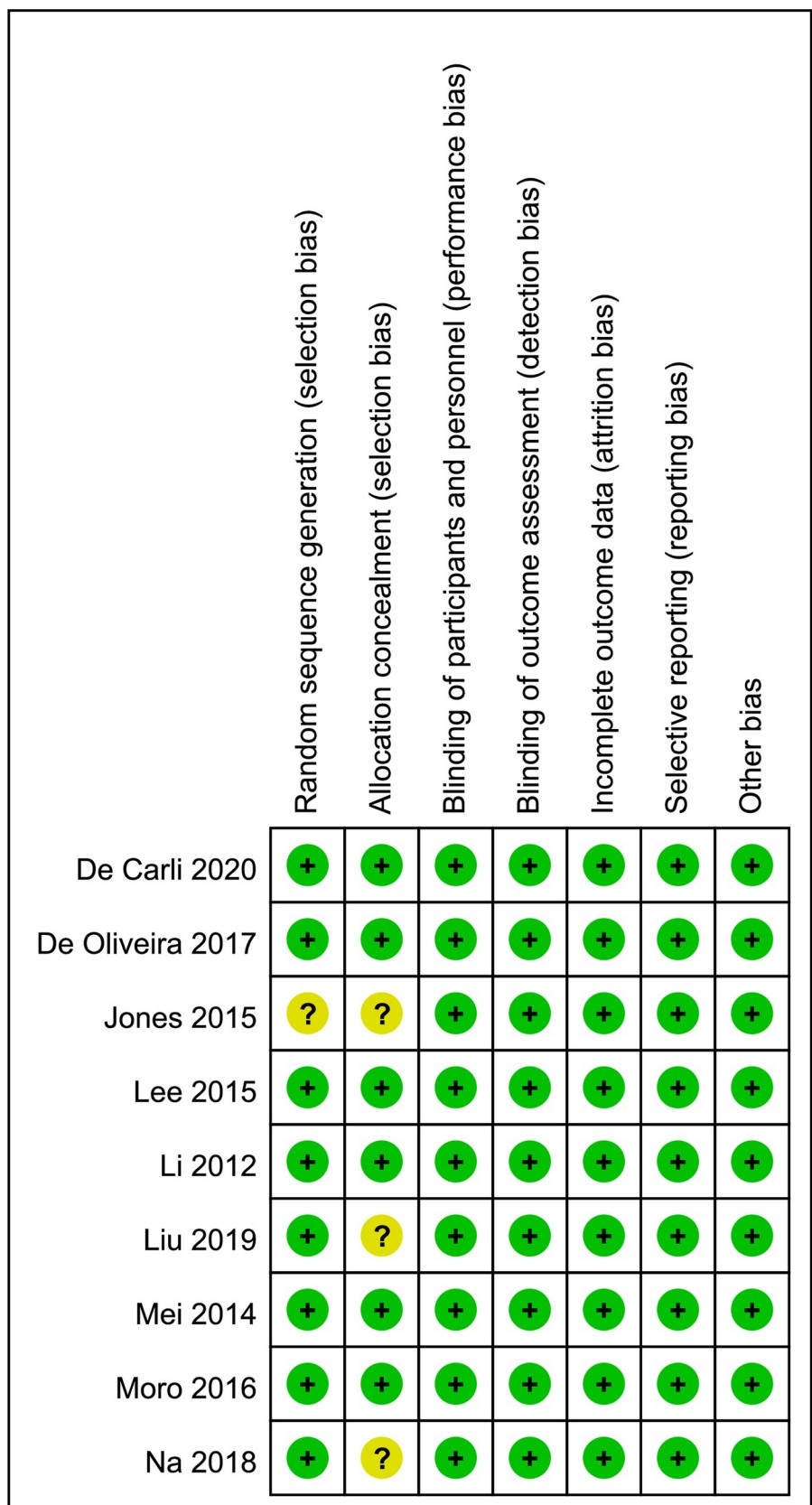

**Fig 2. Risk of bias summary: Review authors' judgements about each risk of bias item for each included study.**

surgery [9–12,18], three on postoperative day 1 (POD1) [3,13,19] and one on postoperative day 2 (POD2) [13], respectively. Data of three studies [3,19,20] were pooled in total QoR-40 score on the day of surgery. Data of eight studies [3,9–13,18,19] were pooled in total QoR-40 score on POD1. Six studies [3,9,11–13,20] reported scores of five dimensions respectively. Data of five studies [3,9,11–13] were pooled in scores in POD1.

Meta-analysis revealed a significant increase in total QoR-40 score on the day of surgery with TIVA compared to inhalational maintenance (MD, 5.91 points; 95% CI, 2.14 to 9.68 points; $P = 0.002$; $I^2 = 0.0\%$). There was no significant difference between groups in total QoR-40 score on POD1 ($P = 0.120$). Only one study reported the score on POD2, with no significant difference between groups in original data ($P = 0.056$) [13]. At 6 hours after surgery, TIVA group had higher scores of four dimensions except the "pain". There was no significant difference between groups in score of each dimension on POD1(Figs 3 and 4).

We used the GRADE approach to judge the certainty of the evidence for total QoR-40 score on the day of surgery to be low. The quality of evidence for total QoR-40 score on POD1 was also classified as "low" (S2 and S3 Files).

### 3.5. Sensitivity analysis

We used the alternate meta-analytic effects model for these outcomes. We used a fixed-effect model which did not alter interpretation of all the results. We also conducted a sensitivity analysis to assess the impact of individual trials on the overall results. We excluded each study sequentially in subgroups on POD1. After removing one study (Lee 2015), the effect of physical comfort on POD1 remained the same but statistical heterogeneity was reduced ($I^2 = 0.0\%$). After removing one study (De Carli 2020), the effect of physical independence on POD1 altered, showing an increase in TIVA group (MD, 0.79 points; 95% CI, 0.02 to 1.56 points; $P = 0.040$; $I^2 = 59.0\%$). Other subgroup effects on POD1 did not alter.

### 3.6. Trial sequential analysis

All trials in each outcome were included in the Trial Sequential Analysis. The estimated RIS for total QoR-40 score on the day of surgery was 321, which was not surpassed by recovered evidence in our meta-analysis. The cumulative Z-curve for total QoR-40 score on the day of surgery crossed the TSA monitoring boundary for benefit (Fig 5A). But in penalized test, the adjusted Z-curve (Fig 5B, green line) did not cross the conventional boundary and the TSA monitoring boundary. It indicates that more evidence is needed to support high QoR-40 score on the day of surgery with TIVA.

In Fig 5C and 5D, the estimated RIS for total QoR-40 score on POD1 was 2787, and the Z-curve and adjusted Z-curve did not cross the conventional boundary and the TSA monitoring boundary, which also indicates that more evidence is needed for a firm conclusion.

## 4. Discussion

The systematic review and meta-analysis of nine RCTs evaluated the impact of TIVA versus inhalational maintenance on postoperative recovery quality using QoR-40 questionnaire. It revealed that TIVA might improve QoR-40 score on the day of surgery. The main improvement might be in four dimensions, including "physical comfort", "emotional status", "psychological support" and "physical independence". No significant differences were found in QoR-40 scores on POD1. However, the Trial Sequential Analysis showed that the estimated RIS for total QoR-40 scores were not surpassed by recovered evidence in our meta-analysis. And the adjusted Z-curves did not cross the conventional boundary and the TSA monitoring

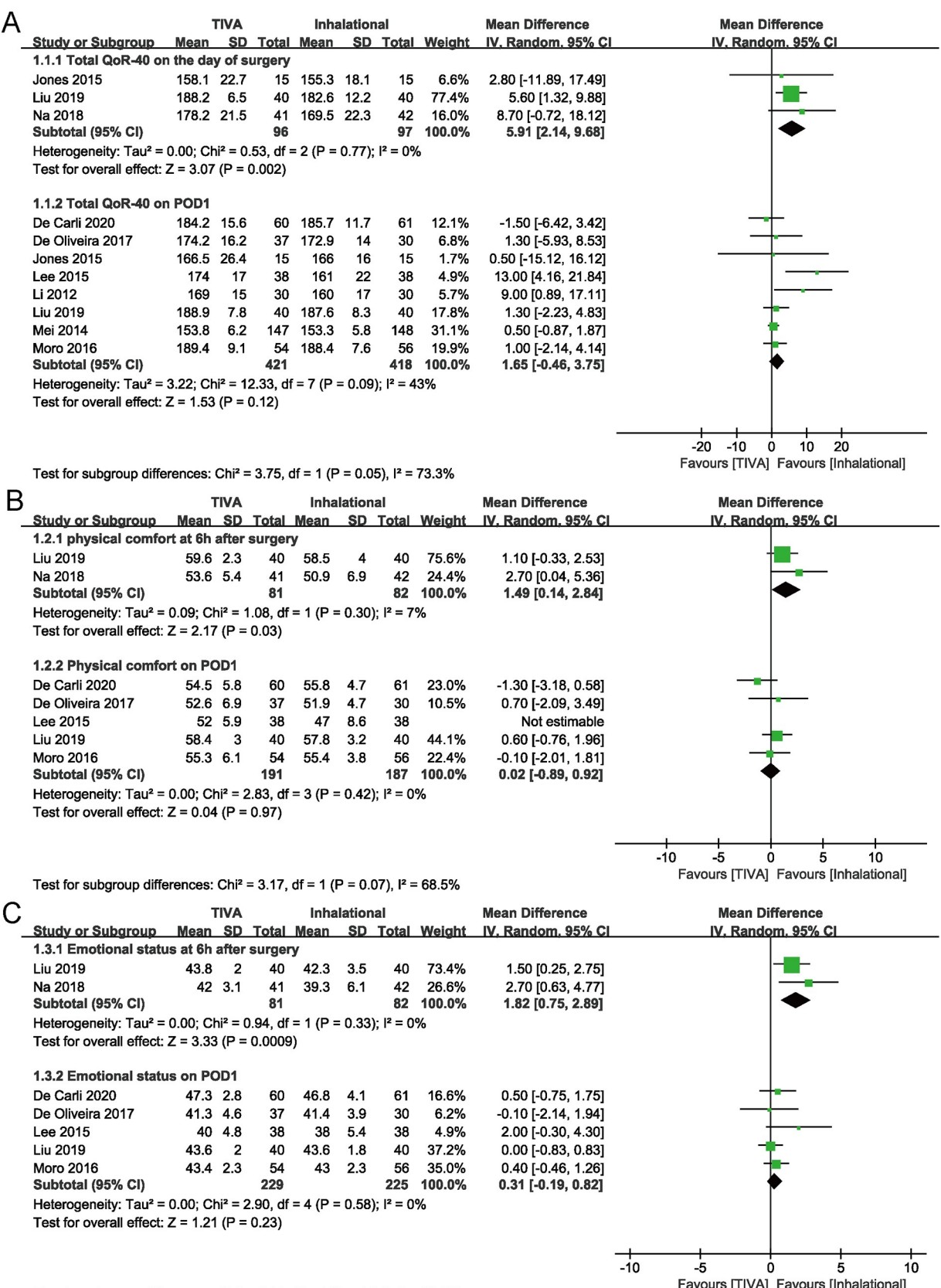

**Fig 3. Forest plots for total QoR-40 (A) and physical comfort (B) with TIVA versus inhalational maintenance.** CI, confidence interval; df, degrees of freedom; $I^2$, heterogeneity index; IV, inverse variance; SD, standard deviation; QoR-40, Quality of Recovery-40 questionnaire; TIVA, total intravenous anesthesia.

boundary. Therefore, more evidence is needed for a firm conclusion, then we need to consider the clinical significance.

We used the GRADE approach and considered study limitations during "Risk of bias" assessment which may influence the certainty of the evidence for each outcome. We excluded one study [21] reporting total QoR-40 scores at 24 hours and 48 hours after surgery because there was no enough information to assess the biases. For the heterogeneities, we attempted to explore but were unable to identify potential explanations. Most of the qualities of evidence were categorized as low according to GRADE system. In sensitivity analysis, when we used a fixed-effect model or excluded each study sequentially, most interpretations of result did not alter. It revealed that the results were stable. The slight alteration appeared in the effects of physical independence on POD1. It might be due to the mean difference (95% CI) was too close to the invalid line. Though it detected a statistically significant increase in QoR score but not reflected the true clinical effect. A 0.79-points increase in TIVA group could not be considered clinically important.

Several comparisons between TIVA and inhalational maintenance of anesthesia have been conducted previously. Miller et al [1] published a meta-analysis about the postoperative cognitive outcomes in elderly people undergoing non-cardiac surgery between the two anesthetic techniques. They found low-certainty evidence that maintenance with propofol-based TIVA may reduce postoperative cognitive dysfunction (POCD). However, the certainty of the evidence in incidences of postoperative delirium, mortality, or length of hospital stay was very low. Herling et al [5] conducted a meta-analysis and compared postoperative pain, postoperative nausea and vomiting (PONV), intraocular pressure (IOP) between the two anesthetic techniques for adults undergoing transabdominal robotic assisted laparoscopic surgery. Due to small number of studies, low-quality evidence suggested that TIVA reduced PONV and prevented an increase in IOP. Another meta-analysis [6] included coronary artery bypass grafting (CABG) patients revealed that there were no significant differences between the volatile anesthetics and TIVA groups in operative mortality, one-year mortality, or any of the postoperative safety outcomes.

These reviews demonstrate that different anesthetic techniques may affect postoperative recovery quality. The QoR-40 questionnaire is an overall assessment of postoperative recovery quality. The validity, reliability, responsiveness, acceptability, feasibility and cross-cultural adaptation have been confirmed [8]. It has been considered as an optimal tool to assess the quality of recovery. This is the first meta-analysis to address patient perception of postoperative quality of recovery of the two anesthetic techniques. But considering the results of trial sequential analysis, our outcomes need more evidence to be confirmed. Combined with previous similar meta-analyses, there is insufficient good quality evidence to conclude that TIVA is better than inhalational maintenance.

There were several limitations to this review: (1) The number of included studies was small (only 9 studies), and most studies had small sample sizes. (2) Different studies reported assessment in different time points, and three studies only reported the total QoR-40 scores, which reduced the number of studies in each subgroup. (3) Most types of surgery involved in these studies were minor surgeries, the results may be different in major surgeries which represent different nociceptive stimuli. (4) Participants of most studies were females, and the conclusion should be cautiously interpreted. (5) The ASA status, mean age, anesthetic management and

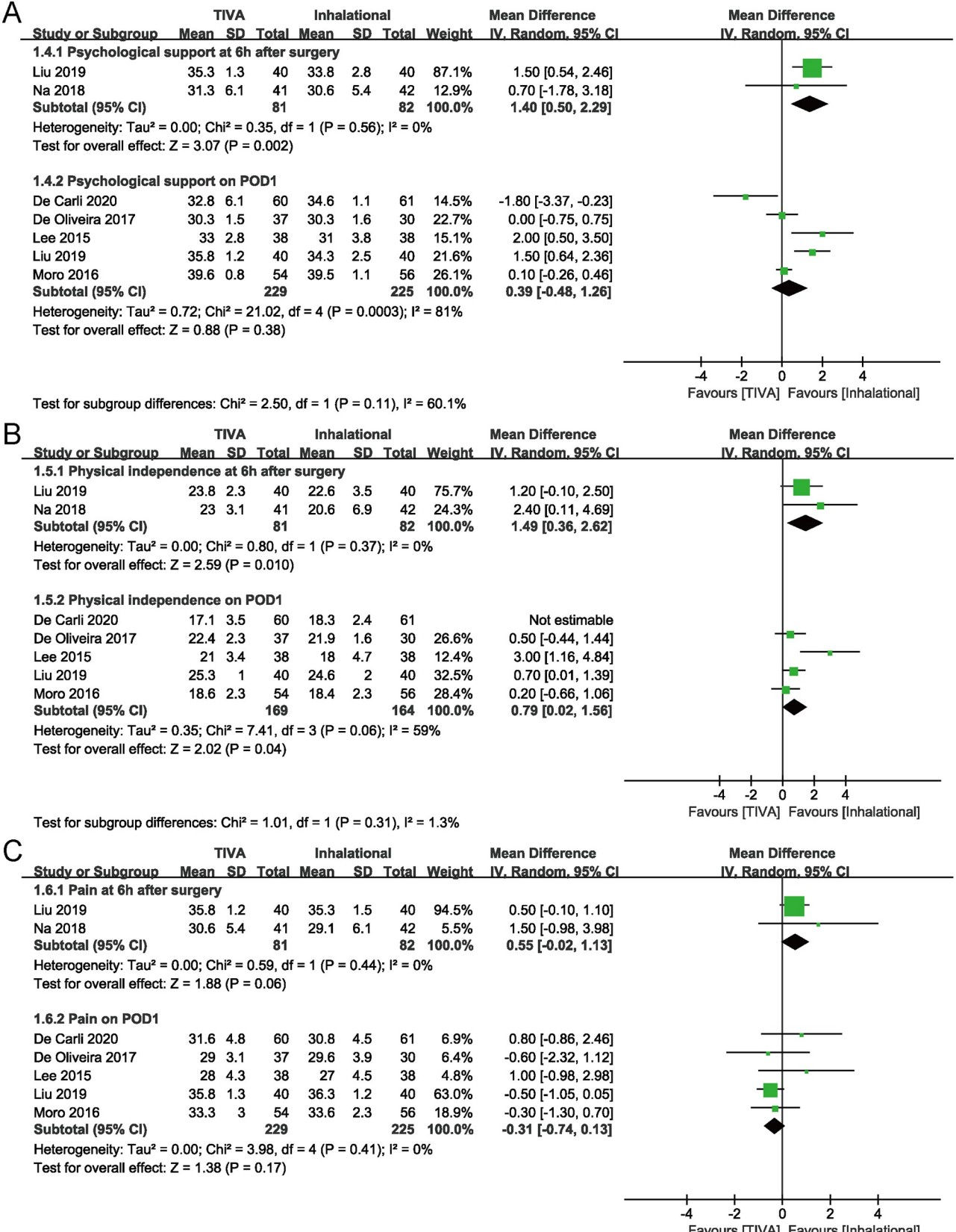

**Fig 4. Forest plots for emotional status (A) and psychological support (B) with TIVA versus inhalational maintenance.** CI, confidence interval; df, degrees of freedom; $I^2$, heterogeneity index; IV, inverse variance; SD, standard deviation; QoR-40, Quality of Recovery-40 questionnaire; TIVA, total intravenous anesthesia.

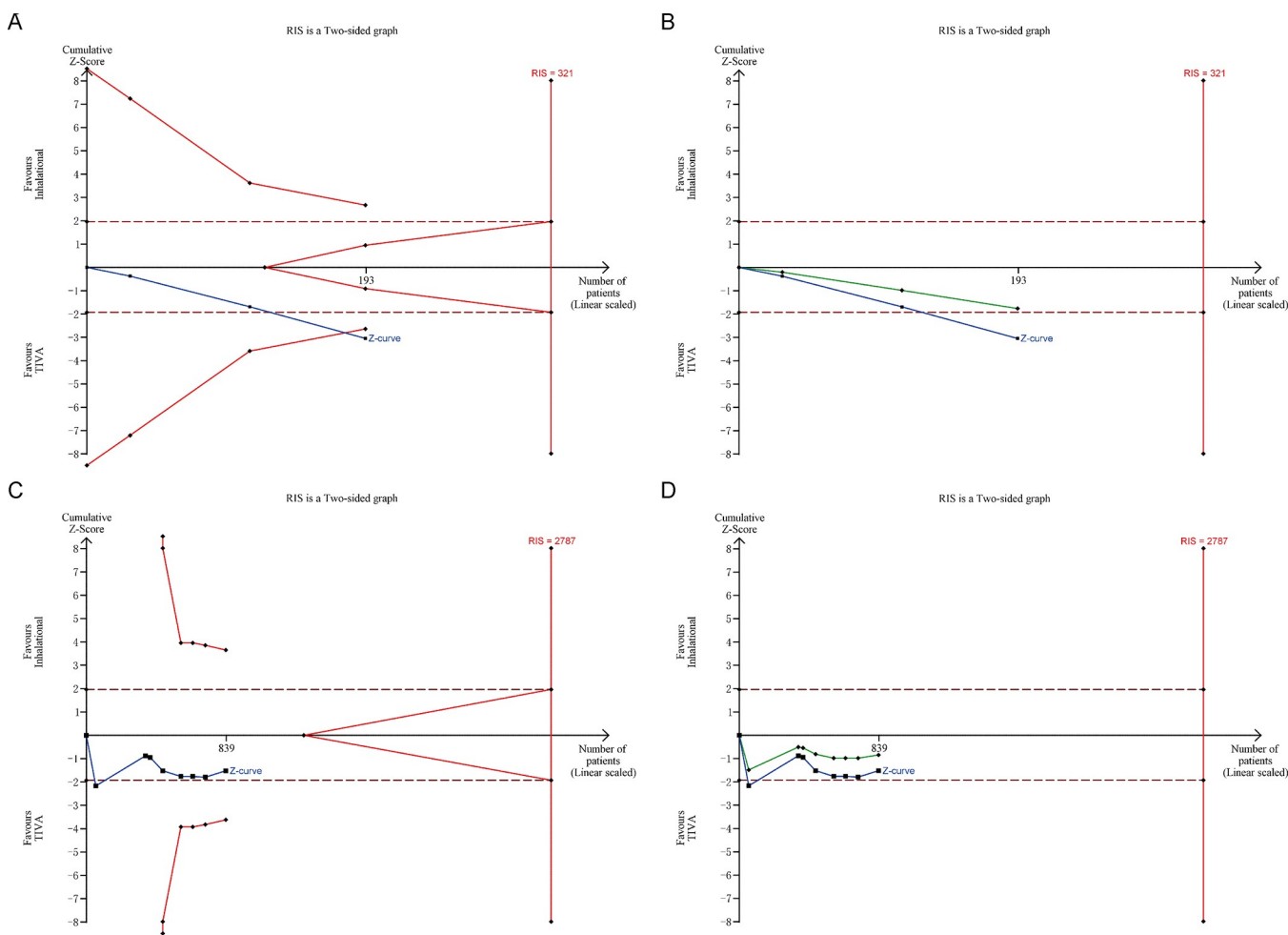

**Fig 5. Forest plots for physical independence (A) and pain (B) with TIVA versus inhalational maintenance.** CI, confidence interval; df, degrees of freedom; $I^2$, heterogeneity index; IV, inverse variance; SD, standard deviation; QoR-40, Quality of Recovery-40 questionnaire; TIVA, total intravenous anesthesia.

monitoring of depth of anesthesia differed between the included studies. These differences may introduce inconsistency and reduce the overall applicability of the evidence.

In conclusion, we found low-certainty evidence that propofol-based TIVA might improve the QoR-40 score on the day of surgery. However, the trial sequential analysis showed that more evidence is needed for a firm conclusion and clinical significance.

## Supporting information

**S1 Checklist. PRISMA 2009 checklist.**
(DOC)

**S1 File. Search strategy.**
(DOCX)

**S2 File. SoF table JCA.**
(DOCX)

**S3 File. Funnel plot.**
(DOCX)

## Author Contributions

**Data curation:** Min Shui, Ziyi Xue.

**Formal analysis:** Min Shui, Xiaolei Miao.

**Methodology:** Anshi Wu.

**Software:** Ziyi Xue.

**Supervision:** Min Shui, Changwei Wei, Anshi Wu.

**Writing – original draft:** Min Shui, Ziyi Xue.

**Writing – review & editing:** Xiaolei Miao, Changwei Wei, Anshi Wu.

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
