## [Decision Letter · Decision Letter 0]

11 Jan 2021

PONE-D-20-35422

Intravenous versus inhalational maintenance of anesthesia for quality of recovery in adult patients: a systematic review with meta-analysis and trial sequential analysis

PLOS ONE

Dear Dr. Wu,

Thank you for submitting your manuscript to PLOS ONE. After careful consideration, we feel that it has merit but does not fully meet PLOS ONE’s publication criteria as it currently stands. Therefore, we invite you to submit a revised version of the manuscript that addresses the points raised during the review process.

We look forward to receiving your revised manuscript.

Kind regards,

Ahmed Negida, MD

Academic Editor

PLOS ONE

Journal Requirements:

3. Please include your tables as part of your main manuscript and remove the individual files. Please note that supplementary tables (should remain/ be uploaded) as separate "supporting information" files.

Reviewers' comments:

Reviewer's Responses to Questions

**Comments to the Author**

1. Is the manuscript technically sound, and do the data support the conclusions?

Reviewer #1: Yes

Reviewer #2: Yes

2. Has the statistical analysis been performed appropriately and rigorously? 

Reviewer #1: Yes

Reviewer #2: Yes

3. Have the authors made all data underlying the findings in their manuscript fully available?

Reviewer #1: Yes

Reviewer #2: Yes

4. Is the manuscript presented in an intelligible fashion and written in standard English?

Reviewer #1: Yes

Reviewer #2: Yes

5. Review Comments to the Author

Reviewer #1: The authors conducted a meta-analysis with trial sequential analysis to determine if total intravenous anesthesia was better than inhalation anesthetics in global recovery of patients. The clinical trials included in this meta-analysis used Quality of Recovery-40 score (QoR 40). The primary studies mostly involved mostly females, patients undergoing outpatient or ambulatory surgeries, intermediate to low risk surgeries. This is an important question for anesthesiologist, which to date has generated inconclusive results. This study attempts to combine results from nine studies and in essence reaches the same conclusion that the results are inconclusive.

General Comments

1. Though the total number of studies is 9 and n=922 patients, in reality there are two types of studies. One group of studies reporting results by six hours which has n of 193 and a second group reporting results by postoperative day one with an n= 839. The smaller group shows some benefit of total intravenous anesthesia, but the larger group does not show a difference in global recovery as reported by by QoR 40 score. Both groups do not achieve RIS threshold, so based on trial sequential analysis one can consider that the meta-analysis is under-powered to date.

2. All studies did not report all domains and sub-analysis of domains contain even less number of patients, which highlight another limitation of the study.

3. Trial Sequential analysis have their own limitations and it may be helpful for readers if the authors highlight those limitations (see Anaesthesia 2020, 75, 15-20) and explain how they have addressed some of the methodological limitation of the study.

4. This study provides a framework for future studies and I suspect, as with many meta-analyses, this is probably one of the initial ones in preparation for other analysis in future.

5. In my opinion considering that the studies showing benefit of total intravenous anesthesia are limited to six hours and have less than 200 patients, it would be advisable to tone down the benefit of total intravenous anesthesia versus inhalational anesthetic and highlight more than methodology, it's limitations and inconclusive nature of this question so far.

6. Another point to highlight is that the difference in absolute score is very small and its clinical significance is not clear.

Specific comments

P4 L71: What is meant by ‘quality of healthcare resources”?

P4 L81: …’discomfort’ sounds very informal. May be authors may consider ‘ …. various factors impact quality of recovery of patients after surgery. ‘

Figure 4: Post-operative Day 1 Physical independence

The total n should be 229 and 225. Please amend.

Reviewer #2: Title:

Intravenous versus inhalational maintenance of anesthesia for quality of recovery in adult patients

Suggested: Intravenous versus inhalational maintenance of anesthesia for quality of recovery in adult patients

undergoing non-cardiac surgery.

Abstract:

Background: Second line: It is still controversial as to which technique

Suggested: It is still controversial which technique

Third line: This meta-analysis aimed to compare impact

Suggested: This meta-analysis aimed at comparing the impact

Methods:

Line 8/9: estimated required information size for total QoR-40 scores were not surpassed by

recovered evidence in our meta-analysis

Suggested: estimated required information size for total QoR-40 scores was not surpassed by

recovered evidence in our meta-analysis.

Page 5

All relevant data are within the manuscript and its Supporting Information files.

Suggested: All relevant data are within the manuscript and its supporting information files

Body of manuscript:

Line 25: It is still controversial as to which technique

Suggested: It is still controversial which technique

Line 26: This meta-analysis aimed to compare impact

Suggested: this meta-analysis aimed at comparing the impact

Line 69: The introduction mentioned (ambulatory anaesthesia) while the title is not about ambulatory

anaesthesia??!

Line 80: Please, omit reference number 7 and omit it from the references list as it is related to paediatric

anaesthesia and the scope of the manuscript is adult anaesthesia.

Line 127: Kindly, (Supplementary) to be written as (supplementary)

Line 132/133:

Here the 9 included studies used the intravenous induction route so the manuscript compares TIVA with

inhalational maintenance, so inhalational induction should be omitted.

Line 258: Kindly, change (Supplementary) to be (supplementary).

Discussion:

Lines 298/299:

more evidence is needed for a firm conclusion. Then we need to consider the clinical significance.

Suggested:

more evidence is needed for a firm conclusion, then we need to consider the clinical significance.

Line 309:

It might due to the mean difference (95% CI) too close to the invalid line

Suggested:

It might be due to the mean difference (95% CI) was too close to the invalid line

Line 333:

meta-analysis to address patient perception of postoperative recovery quality between the two anaesthetic

techniques.

Suggested:

meta-analysis to address patient perception of postoperative quality of recovery of the two anaesthetic

techniques.

Line 344:

participants of most studies were female

Suggested:

participants of most studies were females

Line 354:

And including these studies in future review updates would increase certainity of the effect.

Suggested:

And including these studies in future review updates will increase certainity of the effect.

References:

Reference number 19 link is invalid

6. PLOS authors have the option to publish the peer review history of their article (what does this mean?). If published, this will include your full peer review and any attached files.

Reviewer #1: No

Reviewer #2: No

---

## [Author Response · Author response to Decision Letter 0]

4 Feb 2021

General Comments

1. Though the total number of studies is 9 and n=922 patients, in reality there are two types of studies. One group of studies reporting results by six hours which has n of 193 and a second group reporting results by postoperative day one with an n= 839. The smaller group shows some benefit of total intravenous anesthesia, but the larger group does not show a difference in global recovery as reported by by QoR 40 score. Both groups do not achieve RIS threshold, so based on trial sequential analysis one can consider that the meta-analysis is under-powered to date.

From reported evidence, our meta-analysis and trial sequential analysis, it is true that we couldn’t draw a firm conclusion, but it is real and it can provide reference for future research.

2. All studies did not report all domains and sub-analysis of domains contain even less number of patients, which highlight another limitation of the study.

We couldn’t agree more. These limitations were discussed in our manuscript.

3. Trial Sequential analysis have their own limitations and it may be helpful for readers if the authors highlight those limitations (see Anaesthesia 2020, 75, 15-20) and explain how they have addressed some of the methodological limitation of the study.

It is true that trial sequential analysis (TSA) have their own limitations. The TSA aimed to make the conclusion more reliable. And we complied with its rules. Thus, we think there may be no need to discuss the limitations of TSA in this meta-analysis. 

 4. This study provides a framework for future studies and I suspect, as with many meta-analyses, this is probably one of the initial ones in preparation for other analysis in future.

There seems no requirement for revision in this comment.

5. In my opinion considering that the studies showing benefit of total intravenous anesthesia are limited to six hours and have less than 200 patients, it would be advisable to tone down the benefit of total intravenous anesthesia versus inhalational anesthetic and highlight more than methodology, it's limitations and inconclusive nature of this question so far.

Our meta-analysis and TSA results clearly showed that current evidence couldn’t draw a firm conclusion. The current evidence does not support that total intravenous anesthesia is more advantageous.

6. Another point to highlight is that the difference in absolute score is very small and its clinical significance is not clear.

In our meta-analysis, trial sequential analysis showed that the difference in absolute score was not established, its clinical significance is certainly not clear. Thus, more evidence is needed. That was our conclusion.

Specific comments

P4 L71: What is meant by ‘quality of healthcare resources”?

We revised it in the manuscript.

P4 L81: …’discomfort’ sounds very informal. May be authors may consider ‘ …. various factors impact quality of recovery of patients after surgery. ‘

We revised it in the manuscript.

Figure 4: Post-operative Day 1 Physical independence

The total n should be 229 and 225. Please amend.

The study (De Carli 2020) was not estimable. Thus, the total number is right.

Reviewer #2: Title:

Intravenous versus inhalational maintenance of anesthesia for quality of recovery in adult patients

Suggested: Intravenous versus inhalational maintenance of anesthesia for quality of recovery in adult patients

undergoing non-cardiac surgery.

We revised it in the manuscript.

Abstract:

Background: Second line: It is still controversial as to which technique

Suggested: It is still controversial which technique

We revised it in the manuscript.

Third line: This meta-analysis aimed to compare impact

Suggested: This meta-analysis aimed at comparing the impact

We revised it in the manuscript.

Methods:

Line 8/9: estimated required information size for total QoR-40 scores were not surpassed by

recovered evidence in our meta-analysis

Suggested: estimated required information size for total QoR-40 scores was not surpassed by

recovered evidence in our meta-analysis.

We revised it in the manuscript.

Page 5

All relevant data are within the manuscript and its Supporting Information files.

Suggested: All relevant data are within the manuscript and its supporting information files

We revised it in the manuscript.

Body of manuscript:

Line 25: It is still controversial as to which technique

Suggested: It is still controversial which technique

We revised it in the manuscript.

Line 26: This meta-analysis aimed to compare impact

Suggested: this meta-analysis aimed at comparing the impact

We revised it in the manuscript.

Line 69: The introduction mentioned (ambulatory anaesthesia) while the title is not about ambulatory

anaesthesia??!

Line 80: Please, omit reference number 7 and omit it from the references list as it is related to paediatric

anaesthesia and the scope of the manuscript is adult anaesthesia.

We revised it in the manuscript.

Line 127: Kindly, (Supplementary) to be written as (supplementary)

We revised it in the manuscript.

Line 132/133:

Here the 9 included studies used the intravenous induction route so the manuscript compares TIVA with

inhalational maintenance, so inhalational induction should be omitted.

We revised it in the manuscript.

Line 258: Kindly, change (Supplementary) to be (supplementary).

We revised it in the manuscript.

Discussion:

Lines 298/299:

more evidence is needed for a firm conclusion. Then we need to consider the clinical significance.

Suggested:

more evidence is needed for a firm conclusion, then we need to consider the clinical significance.

We revised it in the manuscript.

Line 309:

It might due to the mean difference (95% CI) too close to the invalid line

Suggested:

It might be due to the mean difference (95% CI) was too close to the invalid line

We revised it in the manuscript.

Line 333:

meta-analysis to address patient perception of postoperative recovery quality between the two anaesthetic

techniques.

Suggested:

meta-analysis to address patient perception of postoperative quality of recovery of the two anaesthetic

techniques.

We revised it in the manuscript.

Line 344:

participants of most studies were female

Suggested:

participants of most studies were females

We revised it in the manuscript.

Line 354:

And including these studies in future review updates would increase certainity of the effect.

Suggested:

And including these studies in future review updates will increase certainity of the effect.

We revised it in the manuscript.

References:

Reference number 19 link is invalid

We coped the link to the browser and opened it. It is right and valid.

---

## [Decision Letter · Decision Letter 1]

30 Apr 2021

PONE-D-20-35422R1

Intravenous versus inhalational maintenance of anesthesia for quality of recovery in adult patients undergoing non-cardiac surgery: a systematic review with meta-analysis and trial sequential analysis

PLOS ONE

Dear Dr. Wu,

Thank you for submitting your manuscript to PLOS ONE. After careful consideration, we feel that it has merit but does not fully meet PLOS ONE’s publication criteria as it currently stands. Therefore, we invite you to submit a revised version of the manuscript that addresses the points raised during the review process.

We look forward to receiving your revised manuscript.

Kind regards,

Ahmed Negida, MD

Academic Editor

PLOS ONE

Reviewers' comments:

Reviewer's Responses to Questions

**Comments to the Author**

1. If the authors have adequately addressed your comments raised in a previous round of review and you feel that this manuscript is now acceptable for publication, you may indicate that here to bypass the “Comments to the Author” section, enter your conflict of interest statement in the “Confidential to Editor” section, and submit your "Accept" recommendation.

Reviewer #2: (No Response)

Reviewer #3: (No Response)

Reviewer #4: All comments have been addressed

2. Is the manuscript technically sound, and do the data support the conclusions?

Reviewer #2: Yes

Reviewer #3: No

Reviewer #4: Yes

3. Has the statistical analysis been performed appropriately and rigorously? 

Reviewer #2: Yes

Reviewer #3: Yes

Reviewer #4: Yes

4. Have the authors made all data underlying the findings in their manuscript fully available?

Reviewer #2: Yes

Reviewer #3: No

Reviewer #4: Yes

5. Is the manuscript presented in an intelligible fashion and written in standard English?

Reviewer #2: Yes

Reviewer #3: No

Reviewer #4: Yes

6. Review Comments to the Author

Reviewer #2: Comments:

1- Line 69 and 70:

Requirement for effectiveness and efficiency of healthcare resources prompts anesthesiologists to consider techniques that provide fast and high quality of recovery.

Suggested (kindly, observe that there are 2 suggested changes in the statement)

The requirement for effectiveness and efficiency of healthcare resources prompts anesthesiologists to consider techniques that provide a fast and high quality of recovery.

2- Page 4 line 80 and 81:

In recent years researchers have recognized the limitations of the fragmentary measures....

Suggested: In recent years, researchers have recognized the limitations of the fragmentary measures...

3- Please, review the tables in page 12 and 13, the reference numbers are different from the numbers of the references written at the end of the manuscript.

4- Line 313/314: It might due to the mean difference (95% CI) was too close to the invalid line

suggested: It might be due to the mean difference (95% CI) was too close to the invalid line

5- Line 323: there is an extra space just before the word Herling, please omit this extra space.

Reviewer #3: Thank you for the opportunity to review the metanalysis entitled " Intravenous versus inhalational maintenance of anesthesia for quality of recovery in

adult patients undergoing non-cardiac surgery: a systematic review with meta-analysis and trial sequential analysis"

1) The premise of the analysis is interesting. However, there are concerning points that may/may not be fixable at this point. Important or possibly contentious question:

2) The included studies differ in the time of outcomes analysis - in the majority - smaller sized studies favored TIVA (are ambulatory/short outcome measure time) in addition represent inadequate work in this domain whereas other relatively larger studies showed “no effect”- plus studies are primarily female gender predominant together making it non-reproducible and more susceptible to selection bias. he problem with this analysis — in theory, you can select the trials you want to include and add them to come up with a number, but the question here is -would the average number apply to these diverse interventions which were compared with user-variable, by using different definitions of outcomes.

3) Introduction: Authors did not justify the rationale of the study. The logic "Wanders" and practically ends in the middle of a thought. Why do authors think that a metanalysis is necessary? Please explain the gaps of knowledge in introduction.

4) Very likely that your analysis is not powered to answer this question. I am sure you will agree that the standards for metanalysis should be higher in such questions due to possible impact on the varying clinical practice and choice of anesthetic agents. The ‘required information size’ necessary to account for heterogeneity and multiple comparisons when the RCTs are added, the necessary TSA boundaries are not reached. Moreover, power should be defined apriori.

5) Please add power analysis and sample size calculation.

6) After reading the conclusion, some may overinterpret the results--therefore avoid making any strong statements and include—"future research is likely to have an important impact."

7) Authors stated the conclusions “strongly” when in fact nothing can be concluded so far for sure. They do have a signal but the results remain inconclusive; considering limitations as discussed above in point 1 and 2. At most the authors should state the “results are inconclusive due to limited power”

8) It would be helpful if you show the results of sensitivity analysis. Please clarify what are duplicate studies?

9) Please include the time period of your search, your last search date, and last metanalysis done.

10) Please name the random effect model used or cite the exact model.

11) Non-cardiac surgeries in title may be a bit misleading. Non-cardiac surgeries can vary a lot with all respect. Authors should consider mentioning something like “Same-day surgeries”

12) Discussion: Similar to introduction. Too wordy and not focused to results. I would suggest reframing it according to the obtained results and just reporting the facts focused on the analysis. Plus, various confusing sentence wording and run-on sentences exist that should be restructured throughout. Awkward phrasing throughout the manuscript detracts from the overall quality of the manuscript. Limitation section should be a thoughtful list that caution your results.

Reviewer #4: The manuscript was good and can be accepted

Introduction was good written

Materials and Methods were good written

Results were clear

Discussion was good written

7. PLOS authors have the option to publish the peer review history of their article (what does this mean?). If published, this will include your full peer review and any attached files.

Reviewer #2: No

Reviewer #3: No

Reviewer #4: **Yes: **Mustafa Abd El Raouf

---

## [Author Response · Author response to Decision Letter 1]

27 May 2021

Reviewer #2: Comments:

1- Line 69 and 70:

Requirement for effectiveness and efficiency of healthcare resources prompts anesthesiologists to consider techniques that provide fast and high quality of recovery.

Suggested (kindly, observe that there are 2 suggested changes in the statement)

The requirement for effectiveness and efficiency of healthcare resources prompts anesthesiologists to consider techniques that provide a fast and high quality of recovery.

We revised it in the manuscript.

2- Page 4 line 80 and 81:

In recent years researchers have recognized the limitations of the fragmentary measures....

Suggested: In recent years, researchers have recognized the limitations of the fragmentary measures...

We revised it in the manuscript.

3- Please, review the tables in page 12 and 13, the reference numbers are different from the numbers of the references written at the end of the manuscript.

We revised it in the manuscript.

4- Line 313/314: It might due to the mean difference (95% CI) was too close to the invalid line

suggested: It might be due to the mean difference (95% CI) was too close to the invalid line

We revised it in the manuscript.

5- Line 323: there is an extra space just before the word Herling, please omit this extra space.

We revised it in the manuscript.

Reviewer #3: Thank you for the opportunity to review the metanalysis entitled " Intravenous versus inhalational maintenance of anesthesia for quality of recovery in

adult patients undergoing non-cardiac surgery: a systematic review with meta-analysis and trial sequential analysis"

1) The premise of the analysis is interesting. However, there are concerning points that may/may not be fixable at this point. Important or possibly contentious question:

2) The included studies differ in the time of outcomes analysis - in the majority - smaller sized studies favored TIVA (are ambulatory/short outcome measure time) in addition represent inadequate work in this domain whereas other relatively larger studies showed “no effect”- plus studies are primarily female gender predominant together making it non-reproducible and more susceptible to selection bias. he problem with this analysis — in theory, you can select the trials you want to include and add them to come up with a number, but the question here is -would the average number apply to these diverse interventions which were compared with user-variable, by using different definitions of outcomes.

We agree with these limitations. We conducted the present meta-analysis also due to these controversies. And we just combined data with similar time point.

3) Introduction: Authors did not justify the rationale of the study. The logic "Wanders" and practically ends in the middle of a thought. Why do authors think that a metanalysis is necessary? Please explain the gaps of knowledge in introduction.

We revised it in the last paragraph in introduction.

4) Very likely that your analysis is not powered to answer this question. I am sure you will agree that the standards for metanalysis should be higher in such questions due to possible impact on the varying clinical practice and choice of anesthetic agents. The ‘required information size’ necessary to account for heterogeneity and multiple comparisons when the RCTs are added, the necessary TSA boundaries are not reached. Moreover, power should be defined apriori.

There seems no requirement for revision in this comment.

5) Please add power analysis and sample size calculation.

The estimated required information size (RIS) in TSA results was sample size calculation.

6) After reading the conclusion, some may overinterpret the results--therefore avoid making any strong statements and include—"future research is likely to have an important impact."

We revised it in the manuscript.

7) Authors stated the conclusions “strongly” when in fact nothing can be concluded so far for sure. They do have a signal but the results remain inconclusive; considering limitations as discussed above in point 1 and 2. At most the authors should state the “results are inconclusive due to limited power”

We revised it in the manuscript.

8) It would be helpful if you show the results of sensitivity analysis. Please clarify what are duplicate studies?

We showed the summarized results and alterations of sensitivity analysis. In the sensitivity analysis, we alternated meta-analytic effects model and excluded each study sequentially in score of each dimension. The results included too much single figures, so we did not upload these figures. If there is a need, we can submit the figures for supplementary files.

9) Please include the time period of your search, your last search date, and last metanalysis done.

It showed in the “Search Strategy and Study Selection” part.

10) Please name the random effect model used or cite the exact model.

We revised it in the manuscript.

11) Non-cardiac surgeries in title may be a bit misleading. Non-cardiac surgeries can vary a lot with all respect. Authors should consider mentioning something like “Same-day surgeries”

 We had considered the problem, but not all studies included in our meta-analysis were ‘same-day surgeries’. When we registered the meta-analysis, we did plan to perform a meta-analysis about non-cardiac surgeries.

12) Discussion: Similar to introduction. Too wordy and not focused to results. I would suggest reframing it according to the obtained results and just reporting the facts focused on the analysis. Plus, various confusing sentence wording and run-on sentences exist that should be restructured throughout. Awkward phrasing throughout the manuscript detracts from the overall quality of the manuscript. Limitation section should be a thoughtful list that caution your results.

We revised it in the manuscript.

---

## [Decision Letter · Decision Letter 2]

9 Jun 2021

PONE-D-20-35422R2

Intravenous versus inhalational maintenance of anesthesia for quality of recovery in adult patients undergoing non-cardiac surgery: a systematic review with meta-analysis and trial sequential analysis

PLOS ONE

Dear Dr. Wu,

Thank you for submitting your manuscript to PLOS ONE. After careful consideration, we feel that it has merit but does not fully meet PLOS ONE’s publication criteria as it currently stands. Therefore, we invite you to submit a revised version of the manuscript that addresses the points raised during the review process.

ACADEMIC EDITOR:  Thank you for your submission. There are some minor questions for you. In the last revision you haven't addressed all the modifications requested. Please provide to make the paper suitable for publication. 

We look forward to receiving your revised manuscript.

Kind regards,

Martina Crivellari

Academic Editor

PLOS ONE

Journal Requirements:

Reviewers' comments:

Reviewer's Responses to Questions

**Comments to the Author**

1. If the authors have adequately addressed your comments raised in a previous round of review and you feel that this manuscript is now acceptable for publication, you may indicate that here to bypass the “Comments to the Author” section, enter your conflict of interest statement in the “Confidential to Editor” section, and submit your "Accept" recommendation.

Reviewer #2: (No Response)

Reviewer #4: (No Response)

2. Is the manuscript technically sound, and do the data support the conclusions?

Reviewer #2: Yes

Reviewer #4: Yes

3. Has the statistical analysis been performed appropriately and rigorously? 

Reviewer #2: Yes

Reviewer #4: Yes

4. Have the authors made all data underlying the findings in their manuscript fully available?

Reviewer #2: Yes

Reviewer #4: Yes

5. Is the manuscript presented in an intelligible fashion and written in standard English?

Reviewer #2: Yes

Reviewer #4: Yes

6. Review Comments to the Author

Reviewer #2: This comment is repeated (written and sent to the author before), as there were 2 modifications included but the author did one only, so please, do the second modifications.

- Line 69 and 70:

Requirement for effectiveness and efficiency of healthcare resources prompts anesthesiologists to consider techniques that provide fast and high quality of recovery.

Suggested (kindly, observe that there are 2 suggested changes in the statement)

The requirement for effectiveness and efficiency of healthcare resources prompts anesthesiologists to consider techniques that provide a fast and high quality of recovery.

We revised it in the manuscript.

Originally in the last revision, I put the same comment, however you responded to one change only which is the addition of (The) to the start of the statement. There was another change which is the addition of (a) before (fast and high quality of recovery.).

Reviewer #4: The manuscript was s good

Introduction was good written and including the aim of the study

Materials and Methods were good written

Results were good written and illustrated

Discussion was good written

7. PLOS authors have the option to publish the peer review history of their article (what does this mean?). If published, this will include your full peer review and any attached files.

Reviewer #2: No

Reviewer #4: No

---

## [Author Response · Author response to Decision Letter 2]

16 Jun 2021

We added (a) before (fast and high quality of recovery).

---

## [Editor Report · Decision Letter 3]

24 Jun 2021

Intravenous versus inhalational maintenance of anesthesia for quality of recovery in adult patients undergoing non-cardiac surgery: a systematic review with meta-analysis and trial sequential analysis

PONE-D-20-35422R3

Dear Dr. Wu,

We’re pleased to inform you that your manuscript has been judged scientifically suitable for publication and will be formally accepted for publication once it meets all outstanding technical requirements.

Kind regards,

Martina Crivellari

Academic Editor

PLOS ONE
---

## [Editor Report · Acceptance letter]

29 Jun 2021

PONE-D-20-35422R3 

Intravenous versus inhalational maintenance of anesthesia for quality of recovery in adult patients undergoing non-cardiac surgery: a systematic review with meta-analysis and trial sequential analysis 

Dear Dr. Wu:

I'm pleased to inform you that your manuscript has been deemed suitable for publication in PLOS ONE. Congratulations! Your manuscript is now with our production department. 

Kind regards, 

on behalf of

Dr. Martina Crivellari 

Academic Editor

PLOS ONE